# Chitinolytic and Fungicidal Potential of the Marine Bacterial Strains Habituating Pacific Ocean Regions

**DOI:** 10.3390/microorganisms11092255

**Published:** 2023-09-08

**Authors:** Iuliia Pentekhina, Olga Nedashkovskaya, Aleksandra Seitkalieva, Vladimir Gorbach, Lubov Slepchenko, Natalya Kirichuk, Anna Podvolotskaya, Oksana Son, Liudmila Tekutyeva, Larissa Balabanova

**Affiliations:** 1Institute of Biotechnology, Bioengineering and Food Systems, Advanced Engineering School, Far Eastern Federal University, 10 Ajax Bay, Russky Island, 690922 Vladivostok, Russia; sasha0788@inbox.ru (A.S.); slepchenko.lubov@gmail.com (L.S.); apodvolot7777@mail.ru (A.P.); oksana_son@bk.ru (O.S.); tekuteva.la@dvfu.ru (L.T.); 2Molecular Biology, Biotechnology and Bioinformatics Center, R&D, Arnika Ltd., Volno-Nadezhdinskoe, 692481 Vladivostok, Russia; 3Laboratory of Marine Biochemistry, G.B. Elyakov Pacific Institute of Bioorganic Chemistry, Far Eastern Branch, Russian Academy of Sciences, Prospect 100-Letya Vladivostoka 152, 690022 Vladivostok, Russia; oned2004@mail.ru (O.N.); vigorbach@bk.ru (V.G.); kirichuk_natalya@bk.ru (N.K.)

**Keywords:** marine chitinolytic bacteria, chitin-degrading enzymes, chitinase activity, antifungal activity, whole-genome sequencing, functional genomics

## Abstract

Screening for chitinolytic activity in the bacterial strains from different Pacific Ocean regions revealed that the highly active representatives belong to the genera *Microbulbifer*, *Vibrio*, *Aquimarina*, and *Pseudoalteromonas*. The widely distributed chitinolytic species was *Microbulbifer* isolated from the sea urchin *Strongylocentrotus intermedius*. Among seventeen isolates with confirmed chitinolytic activity, only the type strain *P. flavipulchra* KMM 3630^T^ and the strains of putatively new species *Pseudoalteromonas* sp. B530 and *Vibrio* sp. Sgm 5, isolated from sea water (Vietnam mollusc farm) and the sea urchin *S. intermedius* (Peter the Great Gulf, the Sea of Japan), significantly suppressed the hyphal growth of *Aspergillus niger* that is perspective for the biocontrol agents’ development. The results on chitinolytic activities and whole-genome sequencing of the strains under study, including agarolytic type strain *Z. galactanivorans* Dji^T^, found the new functionally active chitinase structures and biotechnological potential.

## 1. Introduction

The marine environment presents many species of microorganisms, including marine bacteria, which produce a number of unique enzymes different from the terrestrial ones that have an application in industrial development. The properties of marine bacteria enzymes are salt tolerance, hyperthermostability, barophilicity, and cold adaptability that allows them to be cultured in many condition ranges [1]. In the marine environment, chitin, a β-1,4-linked homopolymer of N-acetylglucosamine (GlcNAc), is the most abundant polymer which is a component of crustacean shells, insect exoskeletons, fungal cell walls, etc. [2,3]. An annual production of chitin in nature is 10^11^ tons, but it is not accumulated in the marine environment because marine bacteria produce chitin-degrading enzymes, particularly, glycoside hydrolyse (GH) chitinases and lytic polysaccharide monooxygenases (LPMO), which are involved in the depolymerisation of chitin [4].

The enzymes with chitinolytic activity have been described mainly in seven glycoside hydrolase (GH) families, namely, GH3, GH18, GH19, GH20, GH23, GH48, and GH84, without obvious sequence similarity, and only a few core regions are structurally conserved in the “lysozyme superfamily” [5]. Among them, chitinases (EC 3.2.1.14) are classified into the families GH18 and 19 according to the carbohydrate-active enzymes (CAZy) database (http://www.cazy.org/, accessed on 20 April 2022). The GH18 chitinases have a catalytic domain with the (β/α)_8_-barrel folds and produce a β-anomer product by the retaining mechanism, while the catalytic domain of GH19 chitinases have a high α-helical content and produce the α-anomer product through the inverting mechanism [6,7,8].

The GH18 chitinases are classified into three subfamilies: A, B, and C, where the subfamily A chitinases contain a small α + β domain inserted between the seventh and eighth β-strands of the (β/α)_8_ barrel of the catalytic domain, while the B and C subfamilies do not have such domain [9]. The distribution of GH18 chitinases is in various organisms, such as bacteria, fungi, viruses, plants, and animals, whereas the GH19 chitinases have been primarily found in plants, nematodes, and bacteria [10].

The GH19 endo-acting chitinases were found in early studies as the plant-derived pathogenesis-related bifunctional lysozymes and are classified into the following classes: the class I chitinases, which contain an N-terminal chitin-binding domain and the GH19 catalytic domain; the class II chitinases, which have the GH19 catalytic domain only; the class II-L chitinases, which lack several loop regions and are related to class II chitinases; and the class IV chitinases, which are similar to class I chitinases, but lack several loops. The importance of GH19 chitinases’ loop structure organization is suggested to be shown the substrate binding. Bacterial GH19 chitinases do not contain loops I, II, V, and the C-terminal loop as was previously reported [5,11]; however, according to the loop structure organization, two GH19 chitinases of *Aeromonas salmonicida* were suggested to be novel types of chitinases due to them containing more loops than known bacterial GH19 chitinases [12]. The GH19 chitinases are described as the enzymes which have an inhibitory activity against fungi [5,10,11,12,13].

The chitin bioconversion by chitinases is boosted with LPMOs, which are related to the auxiliary activity protein structural family AA10. These enzymes have a compact distorted β-sandwich structure, stabilized by loops and helices. Such enzymes fulfil the depolymerisation of chitin by the oxidative cleavage of glycosidic bonds to create the new chain ends for the chitinases’ action. The combination of chitinases and LPMOs in the reaction gives a synergy that increases chitin degradation [14,15].

The newly characterized oxidative chitin utilization pathway in a marine bacterium, *Pseudoalteromonas prydzensis* ACAM 620, initiated by LPMOs in the case of crystalline chitin, differs from the well-known hydrolytic chitin utilization pathway in enzymes, transporters, and regulators [4]. The pathway starts with the LPMO-mediated extracellular breakdown of chitin into C1-oxidized chitooligosaccharides. Then, their terminal residues, 2-(acetylamino)-2-deoxy-D-gluconic acid (GlcNAc1A), are hydrolytically released in the periplasm and converted intracellularly to 2-keto-3-deoxygluconate 6-phosphate, acetate, and NH_3_ via a series of reactions resembling the degradation of D-amino acids rather than other monosaccharides. This pathway was found to exist in many marine Gammaproteobacteria, which enhances the degradation of crystalline chitin, highlighting the importance of these bacteria in the initial degradation of insoluble polysaccharides in the marine environment [4].

The chitin-degrading activity, chitinase systems, and individual chitinase genes were identified and characterized in a number of chitinolytic bacteria isolated from the marine habitats [16,17,18,19,20,21,22,23]. Thus, among 11 strains from the aquatic environment, analysed by chitinase activity, only 3 of them (*Photobacterium galatheae* S2753, *Pseudoalteromonas piscicida* S2040, S2724) showed high activities towards chitin on the plates and against *A. niger* and other tested fungi [16]. Screening of 92 marine bacteria revealed that *Pseudoalteromonas rubra* DSM 6842^T^ was the most active towards different kinds of chitin due to the expression of exochitinases [17]. The exo- and endochitinase activities were observed in *Vibrio* sp., *Shewanella* sp., most of *Aquimarina* spp., *Pseudoalteromonas* sp., and *Microbulbifer* sp., isolated from octocoral and marine sponges [18]. The genes encoding for the protein ChtMB509, ChiC from the strains *Microbulbifer thermotolerans* DAU221 and *Pseudoalteromonas* sp. DL-6, respectively, were cloned to be confirmed to possess the salt-tolerant exochitinase activity [21,22]. Three extra salt-tolerant chitinases Chia4287, Chib0431, and Chib0434 from *Pseudoalteromonas flavipulchra* DSM 14401^T^ were heterologously expressed and characterized as active towards crystalline α-chitin, with an optimum temperature of 45–50 °C and an optimum pH of 7.0–7.5, producing mainly (GlcNAc)_2_ [23].

In the aquatic environment, the utilization of chitin by marine bacteria can occur for their growth and survival in the host, similarly to the pathogenic bacterium *Vibrio cholerae*, which uses the chitin-degrading enzymes for colonization of zooplankton or the human gastrointestinal tract [19]. The chitinase system of the pathogenic bacterium *Aliivibrio salmonicida* causing vibriosis in salmonids has been also characterized, the genome of which contains five disrupted chitinase genes (pseudogenes), a GH18 chitinase, and two LPMOs [20].

The purpose of this research was to screen and select the marine bacterial strains from the Collection of Marine Microorganisms (KMM) (https://kmm644.ru/; http://www.piboc.dvo.ru/tmp/KMM.pdf, accessed on 3 August 2023), which isolated samples from sediment, sea water, and habitats of Pacific Ocean regions, with chitinolytic and/or antifungal activities, for sequencing their whole genomes and mining the genes related to chitinase systems.

## 2. Materials and Methods

### 2.1. Bacterial Strains and Culture Conditions

Bacterial strains used in this research were obtained from the Collection of Marine Microorganisms (KMM), G.B. Elyakov Pacific Institute of Bioorganic Chemistry (https://kmm644.ru; http://www.piboc.dvo.ru/tmp/KMM.pdf). Analysed strains were cultivated in Difco Marine Broth (Difco, Becton, Franklin Lakes, NJ, USA) using 18.7 g per 0.5 L at 30 °C and bacteriological agar-agar (American type) using 4.1 g per 0.25 L for routine cultures. The medium for chitinase activity detection and protein production consisted of Bacto Peptone and East Extract 1 g per 1 L, respectively, a cure sea water, and 0.2% colloidal chitin at 30 °C for 3 days.

### 2.2. Chemicals

Colloidal chitin was prepared from powdered chitin (Sigma-Aldrich, Co., St. Louis, MO, USA), following the methods described by Berger LR and Reynolds DM [24].

### 2.3. PCR Amplification and 16S rRNA Gene Sequencing

The genomic DNA of chitinolytic bacteria was isolated by NucleoSpin^®^ Tissue (Macherey-Nagel GmbH&Co.KG, Düren, Germany). The isolated DNA was used to conduct a polymerase chain reaction (PCR), with universal primers to 16S rRNA gene sequences (BF/20: 5′-AGAGTTTGATCMTGGCTCA-3′; BR2/22: 5′-TACGGTTACCTTGTTACGACTT-3′) [25]. The PCR reaction was carried out using an “Encyclo PCR kit” (Evrogen, Moscow, Russia) in a 50 μL reaction tube. The amplified products were detected by electrophoresis on agarose gel (1.2%, *w*/*v*). The target bands in the agarose gel were cut out and purified using a Wizard SV Gel and Clean-Up Kit (Promega Co., Madison, WI, USA). Sequencing was carried out by SeqStudio Genetic Analyzer (Thermo Scientific, Waltham, MA, USA). Nucleotide sequences of obtained 16S rRNA genes were identified by comparison with the known 16S rRNA gene sequences of type strains available at the EzBioCloud 16 S database (https://wwwezbiocloud.net, accessed on 20 November 2022).

### 2.4. Sequencing and Annotation of Genomic DNA

Genomic DNA of all studied strains, with the exception of typical strains, was extracted using the NucleoSpin Tissue kit (Macherey-Nagel, Düren, Germany) following the manufacturer’s instruction. The quantity and quality of the genomic DNA were measured using DNA gel electrophoresis and an Implen NanoPhotometer^®^ (Implen GmbH, München, Germany). The DNA library preparation was carried out using Illumina DNA Prep, (M) Tagmentation kit (Illumina, San Diego, CA, USA), and whole-genome sequencing was performed subsequently using paired-end runs on an Illumina MiSeq platform with 150 bp read lengths for the Sgm 25/1, Sh 4, Sh 5, 14G-22, 14G-20, and V1SW51 strains and 250 bp read lengths for other strains. The sequence quality was assessed via FastQC version 0.11.9 (FastQC, available online: http://www.bioinformatics.babraham.ac.uk/projects/fastqc/, accessed on 6 April 2022) and reads were trimmed using Trimmomatic version 0.39 [26]. Filtered reads were assembled de novo with SPAdes version 3.13.0 [27]. The draft genomes of the strains were annotated and deposited in the GenBank using the NCBI Prokaryotic Genome Annotation Pipeline (PGAP) [28].

### 2.5. Whole-Genome-Based Analysis of the Marine Chitinolytic Bacterial Strains

The genome-based taxonomy and phylogeny of the strains were conducted with an automated platform on the Type (Strain) Genome Server (TYGS) (Type Strain Genome Server (dsmz.de), accessed on 20 April 2023) [29]. Analysed genomes were compared with annotated TYGS genomes of type strains, which are closely related, using MASH fast genome distance estimation [30]. Finally, 10 type strains with minimum distance were selected, confirmed with the gene sequencing analysis of 16S rRNA from RNAmmer [31] followed by BLAST analysis [32] of 14309 variants of type strains in the database. Exact distance was calculated using Genome BLAST Distance Phylogeny (GBDP) with the coverage algorithm and the d5 distance [33]. All pairwise comparisons among genomes were conducted with GBDP and exact intergenomic distances for phylogenetic analysis. Data of digital DNA–DNA hybridization (dDDH) and confidence intervals were calculated by GGDC 2.1 [33]. Obtained intergenomic distances were used for phylogenetic tree construction with FASTME 2.1.6.1, including SPR [34]. Each branch in the tree was visualized using PhyD3 [35]. A cluster of species based on the genomes of type strains was conducted by the radius of dDDH being 70%, including 13 type strains as was described previously [29]. Subspecies clustering was performed using a dDDH range of 79% [36]. The average of whole-genome sequence identification (ANI) was estimated using ANI ChunLab [37].

### 2.6. Chitinase Activity and SDS–PAGE Assay

Analysed strains were cultivated in liquid medium containing 0.2% colloidal chitin at 30 °C, 150 rpm, for 3 days for the chitinolytic enzymes’ detection in the culture supernatant and on colloidal agar plates at the same temperature. The chitinase activity on agar plates was evaluated over several days by visual inspection. The culture supernatant was separated from cells and debris by centrifugation (6000× *g*, 10 min, 4 °C) and dialyzed against 20 mM sodium phosphate buffer (pH 6.0) at 4 °C overnight. The protein concentration was measured by the Bradford protein assay [38]. The protein solution was analysed by sodium dodecyl sulfate–polyacrylamide gel electrophoresis (SDS–PAGE) as described by Laemmli [39] using 12.5% polyacrylamide gels. Protein was concentrated to 30 µg with 10% trichloroacetic acid prior to SDS–PAGE. 

The chitinase activity detection was carried out in a reaction mixture with a total volume of 600 μL, colloidal chitin (0.1%) as a substrate, crude enzyme (the culture supernatant containing 50 µg of the total protein), and 20 mM sodium phosphate buffer (pH 6.0). The reaction mixture was incubated for 15 min at 37 °C, and chitinase activity measurement was assayed using a modified version of the Schales’ procedure [40] with GlcNAc (Sigma-Aldrich, St. Louis, MO, USA) as the standard.

### 2.7. Confirmation of Chitinase Gene Functional Annotations 

Confirmation of the predicted chitinase gene function was performed by screening their recombinant products’ enzymatic activity with the use of chitinase substrates. 

Recombinant proteins were produced in the *Escherichia coli* strain Rossetta DE3 using the pET-40b(+) vector (Novogen, Waltham, MA, USA). The predicted full-length coding sequences (CDS) encoding for the putative mature chitinolytic enzymes (without signal peptides) in accordance with the annotated function were synthesized by polymerase chain reaction (PCR), using the genomic DNA, gene-specific primers, and Q5^®^ High-Fidelity DNA Polymerase (New England Biolabs (NEB), 240 County Road Ipswich, MA, USA). The gene accession numbers and gene-specific primers, with endonuclease restriction sites and annealing temperatures, are presented in the Results and Discussion. The PCR products were purified via electrophoresis in 1% agarose gel with the use of Cleanup Mini kit (Evrogen, Moscow, Russia) and treated with the restriction enzymes NcoI, SacI, SalI, or XhoI in the optimal buffers (Thermo Fisher Scientific, Waltham, MA, USA) for 3 h at 37 ◦C. The restricted DNA was purified by the Cleanup Mini kit (Evrogen, Moscow, Russia). The pET-40b(+) DNA and insertion gene sequences were ligated in 50 µL of ligation buffer according to the instructions (Thermo Fisher Scientific, Waltham, MA, USA). Then, 10 µL of the reaction mixture was used to transform the competent *E. coli* DH5α cells. The transformed clones were grown on Luria-Bertani (LB) agar containing 50 µg/mL kanamycin. After incubation for 16 h at 37 °C, the clones were PCR-based screened, and the target plasmid DNA was propagated, isolated, and sequenced. 

For heterologous expression, the competent cells of the strain *E. coli* Rosetta (DE3) were transformed by the recombinant plasmids as described above, and cells transformed by the plasmids without inserts were used as a control. Ten recombinant clones were grown in 2 mL of the liquid LB medium containing 25 µg/mL of kanamycin at 180 rpm for 16 h at 37 °C. Then, the cells were placed in a fresh LB medium (20 mL) containing kanamycin (25 µg/mL) and incubated at 37 °C on a shaker at 180 rpm until the optical density at 600 nm was 0.6–0.8. After that, 0.2 mM isopropyl-β-D-thiogalactopyranoside (IPTG) was added to induce the recombinant gene expression, and incubation was continued at 16 °C for 17 h at 180 rpm. Cells were pelleted via centrifugation at 4000× *g* rpm for 15 min at 8 °C, suspended in 10 mL of 0.02 M Na-phosphate buffer (pH 7.0), and subjected to an ultrasonic treatment by Bundeline SONOPULS HD 2070 (Berlin, Germany) to provide a complete release of the recombinant proteins from the *E. coli* periplasmic space.

The suspension was centrifuged at 11,000× *g* rpm for 30 min at 8 °C, the precipitate was discarded, and the chitinase activity was determined in the resulting extract. The chitinase activity was measured by the modified version of the Schales’ procedure as described above [40] and by the colorimetric method using 4-nitrophenyl N-acetyl-β-D-glucosaminide (Sigma-Aldrich, St. Louis, MO, USA) as a substrate. The reaction mixture consisted of 50 µL of the *E. coli* cell lysate containing a recombinant protein, 171 µL of the substrate (1 mg/mL), and 279 µL of the 0.02 M Na-phosphate buffer (pH 7.0). Incubation was carried out at 37 °C, and then 500 µL of 0.4 M NaOH was added (stop reagent). Optical density was measured at 400 nm. The protein concentration (C_protein_, mg/mL) was measured according to the Bradford method [38] using bovine serum albumin (BSA) as a reference. Chitinase activity was calculated with the following formulation: U = (ΣV _reaction mixture + stop reagent_, mL × A_400_)/(18.3 _extintion coefficient_ × t _incubation_,min × V_cell lysate_, mL × C _protein_, mg/mL). One unit (U) of chitinase activity was defined as the amount of the enzyme that releases 1 µM of p-nitrophenol in 1 min under the experimental conditions used. Chitinase activity in each clone with targeted sequences was measured in three biological and technical repeats. The difference between the values did not exceed 5%. The strain *E. coli* Rosetta (DE3) carrying the plasmid pET-40b(+) without the target gene insertions was used as the control.

### 2.8. Antifungal Activity Assay

The antifungal activity of bacterial strains was evaluated on agar plates (Difco Marine Broth, bacteriological agar-agar (American type), 0.2% colloidal chitin) by inhibition of the mycelial extension of the fungal strain *Aspergillus niger* KMM 4797 (Collection of Marine Microorganisms (KMM), G.B. Elyakov Pacific Institute of Bioorganic Chemistry; https://kmm644.ru, accessed on 15 December 2022) using overnight-cultured strains. Three drops of fungi spores were spread on the surface of the agar plate. After the plates’ drying, each bacterial strain was cultivated at the distance of 2.5 cm from the centre of plate, and it was cultured at 30 °C for 1 day, followed by 28 °C for the activity detection. The inhibition of the mycelial extension was determined by visual inspection.

## 3. Results and Discission

### 3.1. Characterization of the Marine Chitinolytic Bacterial Strains

Marine chitinolytic bacteria make a major contribution to chitin degradation in the environment, and the study of their chitinase system, particularly chitin-degrading enzymes, has received great interest by researchers due to the number of completely uncharacterized marine bacteria in nature and the unique properties of their proteins [16,21,22,23,41].

In this research, 66 bacterial strains, deposited in the Collection of Marine Microorganisms (KMM) of G.B. Elyakov Pacific Institute of Bioorganic Chemistry (PIBOC) (https://kmm644.ru/; http://www.piboc.dvo.ru/tmp/KMM.pdf, accessed on 3 August 2023), were used for screening the producers of highly active chitinases due to their previously identified chitinolytic phenotypes (Appendix A). The species identification of the strains was carried out by sequencing the 16S rRNA genes and by the combination of morphological, cultural, and biochemical properties using standard methods (Appendix A). All bacterial isolates were Gram-negative, heterotrophic, mesophilic, and required Na+ ions or sea water for growth. Most of the strains belonged to the class *Gammaproteobacteria* of the phylum *Proteobacteria* (15 strains or 87.5%), and only 2 strains (12.5%) belonged to the class *Flavobacteriia* of the phylum *Bacteroidetes* (Appendix A).

The chitinase activity was detected by culturing the strains on agar plates in the presence of colloidal chitin to evaluate clearing zones caused by the substrate degradation. Among them, 17 strains showed the largest cleared zones; therefore, they were chosen for further investigation, including the strains of putatively new species *Vibrio* spp. and *Pseudoalteromonas* sp. (Figure 1; Table 1).

The 16S rRNA gene sequence comparison of these highly active chitinolytics with the type strains in the EzBioCloud database showed that eight strains, Sgf 25, Sgm 25/1, Sh 1, Sh 2, Sh 3, Sh 4, Sh 5, and 14G-22, belong to the genus *Microbulbifer* (family *Microbulbiferaceae* of the phylum *Proteobacteria*); three strains, Sgm 5, Sgm 22, and 14G-20, are representatives of the genus *Vibrio* (family *Vibrionaceae* of the phylum *Proteobacteria*); the strain V1SW 51 is related to the genus *Aquimarina* (family *Flavobacteriaceae* of the phylum *Bacteroidetes*); and the strain B530 belongs to the genus *Pseudoalteromonas* (family *Pseudoalteromonadaceae* of the phylum *Proteobacteria*) [42]. However, the *Vibrio* and *Pseudoalteromonas* isolates have the plural polymorphic 16S rRNA genes that make their species identification difficult without thorough genome-based or species-specific analysis [43,44]. 

Among gammaproteobacteria in this study, the species *Microbulbifer thermotolerans* was the most numerous (nine strains) including the type strain JCM 14709^T^, isolated from the deep-sea sediment microbial mat of Sagam Bay, Kagoshima, Japan [45] (Figure 1, Table 1). The seven strains *M. thermotolerans* Sgf 25, Sgm 25/1, Sh 1, Sh 2, Sh 3, Sh 4, and Sh 5 were isolated from the grey sea urchin *S. intermedius*, a common inhabitant of Troitsa Bay, Peter the Great Gulf, the Sea of Japan, and one strain, *M. thermotolerans* 14G-22, was from a marine sediment sample, collected from the habitat of the sea urchin [46].

The genus *Vibrio* included two strains, Sgm 5 and Sgm 22, from the sea urchin and one strain 14G-20 from the marine sediment at the site of the sea urchin collection (Figure 1; Table 1). Interestingly, the sea urchin-associated strains of *Vibrio* spp. may potentially represent the new species. Thus, the *Vibrio* sp. strain Sgm 5 showed 99.4% PCR-derived 16S rRNA gene sequence similarity with the *V. hyugaensis* and *V. jasicida* species, and the strain Sgm 22 showed ≥98.9% sequence similarity with the known *V. lentus*, *V. echinoideorum, V. atlanticus*, and *V. tasmaniensis* species according to the EzBiocloud BLAST-based results [42]. Despite the close relationship of the *Vibrio* sp. strain 14G-20 to *V. atlanticus* and *V. tasmaniensis* (99.93%), further genomic studies were required to determine its exact taxonomic position and possibly reclassify closely related species.

Among the chitinolytic bacteria of the genus *Pseudoalteromonas*, the *P. flavipulchra* type strain KMM 3630^T^ (=NCIMB 2033^T^), isolated from the marine water sample collected near Niche, France [47], and the strain *Pseudoalteromonas* sp. B530, isolated from the mollusc farm located in a lagoon of Nha Trang Bay (South China Sea, Viet Nam), were selected (Figure 1; Table 1). The strain *Pseudoalteromonas* sp. B530 showed 99.7% similarity with the PCR-based 16S rRNA gene sequence with *P. piscicida* and *P. flavipulchra*. Determination of the exact taxonomic position of the strain *Pseudoalteromonas* sp. B530 also required further species validation based on the whole-genome sequencing.

The chitinolytic *Flavobacteria* were represented by two strains, V1SW 49^T^ and V1SW 51 (Figure 1; Table 1), which belong to the well-known species *Aquimarina muelleri*, isolated from the sea water of the Amursky Bay, Sea of Japan [48], and one strain of *Zobellia galactanivorans* Dji^T^ [49]. The type strain *Z. galactanivorans* Dji^T^ was the single representative of the genus *Zobellia* with a high chitinolytic potential found in this study (Figure 1; Table 1). *Z. galactanivorans* Dsij^T^, isolated from red seaweed in Roscoff (Brittany, France), has been studied to degrade a large set of complex polysaccharides, such as agars and carrageenans from red algae, as well as alginate, laminarin, and fucoidans from brown algae. Accordingly, it is the model organism to study the polysaccharide utilization loci (PUL)-mediated polysaccharide degradation, complementing the knowledge primarily based on gut *Bacteroidetes*. However, there are still no data about its chitinase activity or any encoded chitinase system [49]. Nevertheless, brown algae with their microbiomes are consumed by sea urchins that may lead to their gut colonization by some alga-associated adaptive bacteria that would be useful for the urchin by their polysaccharide-degrading enzymes and, in turn, to genetic exchange with the urchin-associated microbes [50].

### 3.2. Whole-Genome-Based Identification of the Marine Chitinolytic Strains

The de novo whole-genome sequences of the chitinolytic bacterial strains and their functional annotations are deposited in GenBank under the following accession numbers: JAPHPV000000000, JAPHPW000000000, JAPHPX000000000, JAPHPY000000000, JAPHPZ000000000, JAPHQA000000000, JAPHQB000000000, JAPHQC000000000, JAPHQD000000000, JAPHQE000000000, JAPHQF000000000, JAPHQG000000000, and JAPHQH000000000 (Table 1 and Appendix A).

The marine cosmopolitans, *Vibrio* spp. and *Pseudoalteromonas* sp. B530 containing GC from 43.1 to 45.0%, have the largest genome size (4504–5315 kbp) and were associated with marine invertebrates. The highest number of the RNA genes and pseudogenes (up to 83 in *Vibrio* sp. Sgm 5 = KMM 6832), frameshifts sequences, and the absence of function protein translation were identified in their genomes (Appendix A). This could indicate an active regulation and rearrangement of the genomes under the circumstances of the parental clone isolation apart from the population, which occurs very frequently among the *Vibrionaceae* family [51].

Despite the relatively small genomes (from 3366 to 3400 kbp), the *M. thermotolerans* strains isolated from the sea urchin gonads and digestive organs have the largest content of GC (56.4–56.6%), which is common with free-living microorganisms, the habitat of which is complex and has diverse environmental nutrients, especially exposed to increasing temperatures. These data correlate with the genome data of the type strain *M. thermotolerans* DSM 19189^T^ (GC% = 56.2%), a facultative anaerobe and thermophile (growth range 40–49 °C, pH 5.5–9) isolated from bottom sediments of the Sea of Japan at a depth of 2406 m. Currently, this strain is the only validly described representative of the species *M. thermotolerans* [45]. Probably, this bacterium belongs to heterotrophic saprophytes, which are capable of hydrolysing and utilizing the chitinous shells of dead marine invertebrates as a source of nutrients, as well as for penetration to internal organs.

The aerobic heterotrophic marine bacterial strain *A. muelleri* V1SW 51 (KMM 6556) has the lowest GC content (31.4%) (Appendix A), which is consistent with its reduced growth temperature range (4–34 °C) as in the type strain KMM 6020^T^ (=KCTC 12285^T^=LMG 22569^T^), isolated from the sea water sample of the Amur Bay, the Sea of Japan [48].

The whole-genome-based analysis of the marine chitinolytic strains allowed the species identification and delineation, with the exception of the strains *Pseudoalteromonas* sp. B530, *Vibrio* sp. Sgm 22, and *Vibrio* sp. 14G-20.

According to the TYGS analysis results, *Pseudoalteromonas* sp. B530 (=KMM 6841) possesses the new species-forming gene clusters, which are distantly from *P. piscicida* and *P. flavipulchra*, which is correlated with 16S rRNA gene analysis (Appendix A). Moreover, the dDDH (d4%) values for *Pseudoalteromonas* sp. B530 (=KMM 6841) and closely related species are not more than 63.5% with the type strain *P. flavipulchra* LMG 20361 calculated by TYGS. In comparison, the DNA hybridization degree of the type strains *P. maricaloris* LMG 19692, *P. piscicida* ATCC 15057, *P. galatheae* S4498^T^ and *P. peptidolytica* NBRC 101021, and *Pseudoalteromonas* sp. B530 are 63,1, 61,2, 48,3, and 24%, respectively (Appendix A).

*Vibrio* sp. Sgm22 and 14G-20 were also suggested to belong to the new species clustering with the neighbouring species *V. crassostreae*, *V. celticus*, and *V. coralliirubri* (Appendix A). The dDDH (d4%) values for *Vibrio* sp. Sgm 22 (=KMM 6833) and 14G-20 (KMM = 6839) are 100%, contrarily to the dDDH of 43.6% for the type strain *V. coralliirubri* Corallo1^T^ and the closely related species that confirm their relationship to a new species (Appendix A). The most active chitinolytic strain *Vibrio* sp. Sgm 5 (= KMM 6838) is in the same cluster with *V. inhibens* CECT 7692^T^ and *V. jasicida* CAIM 1864^T^ (Appendix A). The results of the digital DNA hybridization (d4%) of the Sgm 5 with the closely related type strains *V. inhibens* CECT 7692^T^ and *V. jasicida* CAIM 1864^T^ are 79.4% and 78.9% (conditional threshold for the species delineation—70%), respectively, so it could belong it to these species. However, in 2016, *V. inhibens* CECT 7692^T^ was proposed to be reclassified to *V. jasicida* [52]. According to the GenBank (NCBI) record, the average nucleotide identity (ANI) for *Vibrio* sp. Sgm 5 (=KMM 6838) and *V. jasicida* CAIM 1864^T^ (ID in GenBank: GCA_001625175.1) equals 97.74% (conditional threshold for the species delineation—95%) with the genome coverages of 90.93% and 89.38%, respectively. Thus, *Vibrio* sp. Sgm 5 (=KMM 6838) belongs to the *V. jasicida* species (Appendix A).

Based on the whole-genome sequence analysis by TYGS, the strains Sgm 25, Sgm 25/1, Sh 1, Sh 2, Sh 3, Sh 4, Sh 5, and Sgm 14G-22 are related to *Microbulbifer thermotholerans* (Appendix A). The strains *M. thermotolerans* Sgf 25, Sgm 25/1, Sh 1, Sh 2, Sh 3, Sh 4, Sh 5, and Sgm 14G-22 showed from 92.5% (Sgm 25/1 and Sh 1) to 95.1% (Sh 3) dDDH (d4%) with the type strain *M. thermotolerans* DSM 19189^T^ (Appendix A). The Sgf 25 strain with the highest chitinase activity isolated from the sea urchin’s hepatopancreas has a 93.2% similarity degree in terms of the DNA hybridization with the type strain (Appendix A).

The whole-genome sequence analysis with TYGS showed that the strain V1SW51 (=KMM 6556) belongs to the species *Aquimarina muelleri* (Appendix A). The dDDH (d4, %) value for the V1SW51 strain and the type strain *A. muelleri* DSM 19832 is 99.8%.

### 3.3. Chitin-Degrading Activity of the Marine Chitinolytic Strains and the Related Gene Annotations

The results of the screening of the marine bacteria isolated from different habitats of Pacific Ocean regions revealed that the strains belonging to the genera of Gram-negative bacteria *Microbulbifer, Vibrio, Pseudoalteromonas*, and *Aquimarina* possess the highest chitinolytic potential (Figure 1).

The strains of *M. thermotolerans* formed the pure zones of chitin digestion already on the fourth day of cultivation, which continued to increase and reached their maximal sizes on the sixth day, with activity predominance in the strains Sgf 25, Sh 1, Sh 3, Sh 4, and JCM 14709^T^ (Figure 1). Interestingly, such strains as *M. thermotolerans* JCM 14709^T^ and Sgm 25/1 primarily had brown-coloured cells, while the colour of the strain Sh 3 was graduated from white to brown (Figure 1). Paulsen et al. (2019) found that the chitin degradation process is a frequent trait for pigmented strains, but it was related to the *Pseudoalteromonas* species. Moreover, their genomes encode the GH19 chitinases more than the non-pigmented strains [41].

In the liquid culture, the strain *M. thermotolerans* Sgf 25 also showed a higher level of chitin-degrading enzymes’ induction than other strains, when grown in the presence of 0.2% colloidal chitin (Figure 2).

Thus, both plated cells and liquid cultures of *M. thermotolerans* showed strong strain-dependent chitinolytic activity towards colloidal chitin, probably due to the different numbers of the encoded or expressed chitin-degrading enzymes (Figure 1 and Figure 2). The liquid cultures of the *M. thermotolerans* strains Sgm 25/1, Sh 5, and Sh 2 showed 4%, 9%, and 13% of the activity in the strain *M. thermotolerans* Sgf 25, respectively, while they all were higher than in the strain Sh 3 (Figure 2). However, the strains Sgf 25, Sh1, JCM 14709^T^, Sgm 25/1, and 14G-22 accordingly contained a significant level of the additional chitin-induced proteins in comparison with their controls, while the distribution of extracellular proteins in SDS–PAGE for the strains Sh 2, Sh 3, and Sh 5 was almost equal to their controls (Figure 3).

Despite the different expression levels of chitin-induced and chitin-degrading proteins in the *M. thermotolerans* strains (Figure 2 and Figure 3), their whole-genome sequences contain the same number of glycoside hydrolase enzymes (4 GH18) and LPMOs (2 AA10), with the exception of the additional third LPMO in the strains Sgm 25/1 and 14G-22 according to the NCBI annotation with the use of the Prokaryotic Genome Annotation Pipeline (PGAP) (Table 1). However, a dbCAN version also suggested three LPMOs in the strains Sh 1, Sh 2, and Sh 5 due to the paralogous AA10 with the domain of carbohydrate-binding motif CBM2. The AA10 protein with CBM73 [53] was found in each strain of *M. thermotolerans* (Appendix A). LPMOs have an important role in chitin degradation, especially in synergy with chitinases, and could increase powdered chitin digestion more than twice compared to the mixture of chitinases only [12,14,53]. According to CAZy, the referent strain *M. thermotolerans* DAU221 also possesses four chitinases of the GH18 family (115 kDa, 63 kDa, 58 kDa, 56 kDa) and two LPMOs (59 kDa and 53 kDa) of the AA10 family. One of the *M. thermotolerans* DAU221 chitinolytic enzymes, MtCh509 (56 kDa), was cloned and characterized as an endochitinase with transglycosylation activity [21].

Among *Vibrio* spp., the strain Sgm 5 showed the visible zone of chitinase activity on the agar plates on the fourth day. However, the strains *Vibrio* sp. 14G-20 and Sgm 22 slightly expanded the activity zones on the sixth day (Figure 1). The liquid extracellular fraction of *Vibrio* sp. Sgm 5 also contained chitinase activity that was 26 and 45% higher in contrast to the strains *Vibrio* sp. Sgm 22 and 14G-20, respectively (Figure 2). Evidently, colloidal chitin induced the expression of the additional extracellular proteins in *Vibrio* sp. Sgm 5, while the strains 14G-20 and Sgm 22 showed no considerable difference with their controls (Figure 3). According to the genome annotations, the equal numbers of GH18 (2) and GH19 (1) were found for all the *Vibrio* strains, with exception of the Sgm 5 genome differing in an additional LPMO (Table 1). The non-identical levels of chitinase production were also observed in the culture supernatant of the *V. parahaemolyticus* RIMD2210633 cells, grown with supplementation of different inducing sugars. *V. parahaemolyticus* produces numerous proteins involved in chitin utilization and expression outside of the cells, not only chitinases for the (GlcNAc)_n_ digestion but also a chitin oligosaccharide deacetylase (COD) for the (GlcNAc)_2_ conversion into GlcNAc-GlcN [54].

The strain *Pseudoalteromonas* sp. B530 showed a high chitinase activity on the fourth day of solid-state cultivation compared to *P. flavipulchra* KMM 3630^T^, but the degradation zones of both strains reached almost the equal sizes on the sixth day (Figure 1). However, the extracellular chitinase activity in the liquid-cultured *P. flavipulchra* KMM 3630^T^ was about two times higher than that in *Pseudoalteromonas* sp. B530 (Figure 2). Nonetheless, the chitin supplementation induced in both *Pseudoalteromonas* sp. B530 and *P. flavipulchra* KMM 3630^T^ had additional extracellular protein production with a molecular weight of 75–100 kDa compared to the control (Figure 3). *P. flavipulchra* KMM 3630^T^ is known to possess three chitinases and two LPMOs, while *Pseudoalteromonas* sp. B530 has three GH18 chitinases, one GH19 chitinase, and two AA10 proteins (Table 1), which could be expressed for chitin digestion in the presence of the inductor (Figure 3). Chitinases Chia4287, Chib0431, and Chib0434 of *P. flavipulchra* DSM 14401^T^ were suggested to be important in degradation of crystalline chitin into (GlcNAc)_2_ [23]. Enzymatic hydrolysis of the recalcitrant chitin by *Pseudoalteromonas* strains has been previously described [22,41]. Thus, *Pseudoalteromonas* sp. DL-6 has two characterized chitinases: a non-processive endo-type chitinase—ChiA—and a processive exo-type chitinase—ChiC—which act synergistically for efficient chitin degradation [22]. The expression of five GH18s, one GH19s, LPMO, and two GH20s was detected in *P. rubra* S4059 by the analysis of its secondary metabolite profiles [55]. Moreover, the newly characterized oxidative chitin utilization pathway in the marine bacterium *P. prydzensis* ACAM 620, initiated by LPMOs towards the crystalline chitin, differs from the well-known hydrolytic chitin utilization pathway in enzymes, transporters, and regulators [4]. In particular, GlcNAc1A is converted to 2-keto-3-deoxygluconate 6-phosphate, acetate, and NH_3_ via a series of reactions resembling the degradation of D-amino acids rather than other monosaccharides. This pathway was found to exist in many marine Gammaproteobacteria, which enhances the degradation of crystalline chitin, highlighting the importance of these bacteria in the initial degradation of crystalline chitin in marine environments [4].

The twofold difference in the chitin-degrading activity was detected in the liquid cultures of *A. muelleri* V1SW 51 and V1SW 49^T^, whereas their visible chitinase activity on the chitin-containing agar was observed only after 11 days of cultivation (Figure 1, Figure 2 and Figure 3). Results of SDS-PAGE revealed that two strains expressed several proteins of 50–75 kDa under colloidal chitin cultivation. The whole-genome analysis showed that the V1SW 51 strain possesses two GH18 chitinases (MCX2762229, MCX2764142) and one LPMO (MCX2761469) (Table 1). V1SW 49^T^ has the same enzyme distribution in the genome (GH18: WP_027414378, WP_081414620; LPMO: WP_051316619). The chitinase activity on agar plates as well as in the culture supernatant of the *Aquimarina* sp. strains was previously reported [18,48]. Interestingly, most *Aquimarina* sp. strains digested colloidal chitin through the expression of extracellular endo- and exochitinases of non-A- and C-types; therefore, the analysed bacteria were suggested to have the novel chitinolytic enzymes. In addition, the functional metagenomics revealed differential chitin degradation and utilization features across free-living and host-associated marine microbiomes [18].

The cleared zones in the plated *Z. galactanivorans* Dji^T^ appeared on the 4th day and slightly increased when the strain was cultivated for 11 days (Figure 1). However, such activity of this agarolytic bacterium was rather towards agar than colloidal chitin, because the type strain *Z. galactanivorans* Dji^T^ is characterized as an efficient degrader of algal biomass [56]. Nevertheless, the *Z. galactanivorans* Dji^T^ total extracellular proteins from the liquid culture showed 53% higher activity towards colloidal chitin relatively to the *A. muelleri* V1SW 49^T^ activity and 7% lower activity than in *A. muelleri* V1SW 51 (Figure 2). Remarkably, a lot of additional proteins were expressed by *Z. galactanivorans* Dji^T^ in response to the supplementation of 0.2% chitin into the liquid nutrient medium (Figure 3). According to Cazy, the type strain Dji^T^ has two chitinases of family 18 (CAZ95072 (~55.5 kDa) and CAZ97205 (~46 kDa)) but no family 19 chitinases or AA10 proteins. However, three lysozymes of GH23 were found in the strain Dji^T^, which may be also responsible for the chitinolytic activity [57]. Barbeyron et al. (2016) have reported about 50 PULs that are encoded in the genome of this bacterium that could be involved in different polysaccharide degradation processes in the environment [58]. However, the chitin-degrading system for the algal polysaccharide-degrading bacterium *Z. galactanivorans* has not yet been characterized. Understanding the role of suggested chitin-degrading enzymes produced by the *Flavobacteria*, *A. muelleri* and *Z. galactanivorans*, will be addressed in further research due to their interesting behaviour against chitin and their completely uncharacterized chitinase systems [18,58].
microorganisms-11-02255-t001_Table 1Table 1Phenotype and genome-based characterization of the chitinolytic potential in marine bacterial strains used in this research.№KMM *Chitinolytic Activity on Agar PlatesFungicidal Activity on Agar PlatesPhylumStrainIsolation Source16S rRNA GeneAccessionNumbersGenome Accession NumbersGenome Size, kbpGC,%CAZyme FamilyChitinase Gene AccessionNumberReference1.–+++−*Proteobacteria**Microbulbifer thermotolerans* JCM 14709^T^A deep-sea sediment in Suruga Bay and Sagami Bay and off Kagoshima, Japan.AB124836CP014864393856.5GH18 (3)SFC67996[45]SFC69774SFC69838AA10 (1)SFC521912.6262+++−*Proteobacteria**Microbulbifer thermotolerans* Sgf 25At a depth of 3 m in Troitsa Bay, Gulf of Peter the Great, Sea of Japan, Pacific Ocean. Host—sea urchin *Strongylocentrotus intermedius*OL744412JAPHQH000000000336656.6GH18 (4)MCX2830300this studyMCX2831405MCX2831407MCX2831458AA10 (2)MCX2831189MCX28322173.6834++−*Proteobacteria**Microbulbifer thermotolerans* Sgm 25/1At a depth of 3 m in Troitsa Bay, Gulf of Peter the Great, Sea of Japan, Pacific Ocean. Host—sea urchin *Strongylocentrotus intermedius*OL744415JAPHQG000000000337456.5GH18 (4)MCX2804080this studyMCX2804660MCX2804711MCX2804713AA10 (3)MCX2805714MCX2806217MCX28062184.6242+++−*Proteobacteria**Microbulbifer thermotolerans* Sh 1At a depth of 3 m in Troitsa Bay, Gulf of Peter the Great, Sea of Japan, Pacific Ocean. Host—sea urchin *Strongylocentrotus intermedius*OL744416JAPHQF000000000336656.5GH18 (4)MCX2841098this studyMCX2842232MCX2842283MCX2842285AA10 (2)MCX2840086MCX28423585.6835++−*Proteobacteria**Microbulbifer thermotolerans* Sh 2At a depth of 3 m in Troitsa Bay, Gulf of Peter the Great, Sea of Japan, Pacific Ocean. Host—sea urchin *Strongylocentrotus intermedius*OL744417JAPHQE000000000338656.4GH18 (4)MCX2834787this studyMCX2835292MCX2835294MCX2835346AA10 (2)MCX2833669MCX28354706.6836+++−*Proteobacteria**Microbulbifer thermotolerans* Sh 3At a depth of 3 m in Troitsa Bay, Gulf of Peter the Great, Sea of Japan, Pacific Ocean. Host—sea urchin *Strongylocentrotus intermedius*OL744418JAPHQD000000000333756.6GH18 (4)MCX2793661this studyMCX2795686MCX2795737MCX2795739AA10 (2)MCX2794997MCX27954217.6837+++−*Proteobacteria**Microbulbifer thermotolerans* Sh 4At a depth of 3 m in Troitsa Bay, Gulf of Peter the Great, Sea of Japan, Pacific Ocean. Host—sea urchin *Strongylocentrotus intermedius*OL744419JAPHQC000000000340056.6GH18 (4)MCX2781498this studyMCX2781500MCX2781953MCX2784221AA10 (2)MCX2782412MCX27830478.6838++−*Proteobacteria**Microbulbifer thermotolerans* Sh 5At a depth of 3 m in Troitsa Bay, Gulf of Peter the Great, Sea of Japan, Pacific Ocean. Host—sea urchin *Strongylocentrotus intermedius*OL744420JAPHQB000000000338256.6GH18 (4)MCX2800845this studyMCX2802019MCX2802021MCX2802072AA10 (2)MCX2800565MCX28027239.6840++−*Proteobacteria**Microbulbifer thermotolerans* 14G-22A marine sediment sample, at a depth of 3 m in Troitsa Bay, Gulf of Peter the Great, Sea of Japan, Pacific Ocean. Host—sea urchin *Strongylocentrotus intermedius*OL744423JAPHQA000000000337556.5GH18 (4)MCX2778775this studyMCX2778989MCX2778991MCX2779042AA10 (3)MCX2778112MCX2778113MCX277938410.6832++++++*Proteobacteria**Vibrio* sp. Sgm 5At a depth of 3 m in Troitsa Bay, Gulf of Peter the Great, Sea of Japan, Pacific Ocean. Host—sea urchin *Strongylocentrotus intermedius*OL744413JAPHPY000000000531545.0GH18 (2)MCX2790314this studyMCX2792044GH19 (1)MCX2791380AA10 (2)MCX2792547MCX279284811.6833++−*Proteobacteria**Vibrio* sp. Sgm 22At a depth of 3 m in Troitsa Bay, Gulf of Peter the Great, Sea of Japan, Pacific Ocean. Host—sea urchin *Strongylocentrotus intermedius*OL744414JAPHPX000000000452844.4GH18 (2)MCX2775709this studyMCX2777646GH19 (1)MCX2774415AA10 (1)MCX277491112.6839++−*Proteobacteria**Vibrio* sp. 14G-20A marine sediment sample, at a depth of 3 m in Troitsa Bay, Gulf of Peter the Great, Sea of Japan, Pacific Ocean. Host—sea urchin *Strongylocentrotus intermedius*OL744424JAPHPW000000000450444.3GH18 (2)MCX2758330this studyMCX2760626GH19 (1)MCX2758955AA10 (1)MCX275781013.6020^T^+−*Bacteroidetes**Aquimarina muelleri* V1SW 49^T^A sea water sample collected in Amursky Bay, Gulf of Peter the Great, Sea of JapanAY608406BMWS00000000495831.0GH18 (2)WP_027414378[48]WP_081414620LPMO (1)WP_05131661914.6556+−*Bacteroidetes**Aquimarina muelleri* V1SW 51A sea water sample, Amursky Bay, Gulf of Peter the Great, Sea of Japan, RussiaOL744421JAPHPZ000000000415231.4GH18 (2)MCX2762229this studyMCX2764142AA10 (1)MCX276146915.3630^T^+++++*Proteobacteria**Pseudoalteromonas flavipulchra* NCMB 2033A sea water sample collected in the Mediterranean Sea, Nice, FranceAF297958AQGY00000000544143.0GH18 (3)MBE0371971[47]MBE0375554MBE0375556LPMO (2)MBE0375052MBE037555516.6841++++++*Proteobacteria**Pseudoalteromonas* sp. B530A sea water sample, a mussel farm located in a lagoon of Nha Trang Bay, South China Sea, Viet NamOL744422JAPHPV000000000472043.1GH18 (3)MCX2765851this studyMCX2765853MCX2769256GH19 (1)MCX2767687AA10 (2)MCX2765852MCX276708117.–+−*Bacteroidetes**Zobellia galactanivorans* Dji^T^A red seaweed in Roscoff (Brittany, France)FP476056FP476056552242.5GH18 (2)CAZ95072[58]CAZ97205GH23CAZ95778CAZ94149* KMM, the Collection of Marine Microorganisms (KMM), G.B. Elyakov Pacific Institute of Bioorganic Chemistry. −, no activity; +, weak activity; ++, slight activity; +++, high activity.

### 3.4. Chitin-Degrading Activity of the Recombinant Proteins Derived from the Predicted Genes of the Marine Chitinolytic Strains

Functional genomics techniques, such as heterologous expression and production of recombinant proteins in *E. coli* cells, were applied to the most active chitinolytic *M. thermotolerans* strain Sgf 25 and the algal polysaccharide-degrading agarolytic *Z. galactanivorans* Dji^T^ to confirm the functional activity of their genes with the predicted chitinase function (Table 2). The expression plasmid pET40b(+) was selected because it equips a signal peptide and a chaperone DsbC appropriate for the E. coli system to produce periplasmic recombinant proteins in the active soluble form [59]. The open reading frame (ORF) sequences, which were predicted to belong to chitinase systems and correspond to the mature proteins with putative chitinase function (Table 1 and Table 2), were taken from the NCBI genome functional annotations deposited in the GenBank or CAZy databases (Appendix A). The chitinase genes encoding for GH18 were isolated by a polymerase chain reaction (PCR), with the use of gene-specific forward and reverse primers, targeted to the N- and C-ends of the genes, respectively, and the chromosomal DNA of the bacterial strains *M. thermotolerans* Sgf 25 and *Z. galactanivorans* Dji^T^ (Table 2). In *Z. galactanivorans* Dji^T^, several genes encoding for GH23 were also chosen for screening chitinase activity due to the chitinase-like function discovery in some GH23 lysozymes and lytic transglycosylases [60,61].

The chitinase activity against colloidal chitin has been determined for the GH18 proteins, namely, 531 Sgf 25, 509 Sgf 25, 571 Sgf25, 1036 Sgf25, and 440 Dji^T^ derived from the predicted chitinase genes of *M. thermotolerans* Sgf25 and *Z. galactanivorans* Dji^T^, respectively (Table 2; Figure 4). Remarkably, the recombinant product 547 DjiT from the gene of *Z. galactanivorans* Dji^T^, which was predicted to belong to the GH23 family [57], also catalysed the reaction of colloidal chitin degradation, with the specific activity approximately equal to the activity of 509 Sgf 25 GH18 (Figure 4).

The reducing sugar ends were hydrolysed from chitin with 10-fold efficiency after the action of the recombinant protein 571 Sgf 25 GH18 (MCX2831407) from *M. thermotolerans* Sgf 25. (Table 2, Figure 4). However, both GH18 proteins 527 Dji^T^ and 440 Dji^T^ from *Z. galactanivorans* Dji^T^ were more active towards 4-nitrophenyl N-acetyl-β-D-glucosaminide comparatively to the GH18 proteins of *M. thermotolerans* Sgf 25 (Table 2), indicating the presence of both endo- and exochitinase activity [62].

### 3.5. Antifungal Activity of the Marine Chitinolytic Strains

Among the chitinolytic marine bacteria under this study, the strains *P. flavipulchra* KMM 3630^T^, *Pseudoalteromonas* sp. B530, and *Vibrio* sp. Sgm 5 only possessed a fungicidal activity against the *Aspergillus niger* strain KMM 4797, revealed by the plate method (Figure 5).

Accordingly, the antifungal activity of *P. flavipulchra* KMM 3630^T^, *Pseudoalteromonas* sp. B530, and *Vibrio* sp. Sgm 5 was expressed as a cleared zone of the *A. niger* 63–144 growth inhibition on the third day of cultivation, which was of approximately 6 mm (Figure 5), whereas other marine chitinolytic strains, including the highly active chitinolytics *M. thermotolerans* Sgf 25 (Figure 1 and Figure 2), did not show any antifungal activity, probably due to the absence of the GH19 enzymes (Table 1). The *P. flavipulchra* KMM 3630^T^, *Pseudoalteromonas* sp. B530, and *Vibrio* sp. Sgm 5 genomes each contain one gene encoding for the GH19 family structure (Table 1). Paulsen et al. (2016) reported that *P. piscicida* S2040, isolated from the Indian Ocean, possessed stronger activity against *A. niger* than the strain S2724 Isolated from the Solomon Islands, and both have several chitinases of GH18 and one GH19 [16]. The GH19 chitinases are commonly known as the enzymes with antifungal activity for biocontrol development [11,13,63,64]. Watanabe et al. (1999) analysed a recombinant chitinase C of the GH19 family, which was first found in a procaryotic organism, *Streptomyces griseus* HUT6037, and showed that this enzyme inhibited growth of *Trichoderma reesei* [63]. Tsujibo et al. (2000) characterized two GH19 chitinases, Chi25 and Chi35, from *S. thermoviolaceus* OPC-520, where Chi35 had stronger antifungal activity against *T. reesei* than Chi25 [64]. Not only do GH19 chitinases have activity against the fungus, but GH18 chitinases are also proposed to have an antifungal activity like Chi18bA from *S. coelicolor* A3(2), which slightly inhibited the growth of several fungi [65]; however, in our research, GH18 did not show such activity. The GH19 chitinases were also reported as the enzymes for an insoluble chitin degradation and (GlcNAc)_2_ production that is important for the efficient chitin bioconversion by the different types of GHs [66].

## 4. Conclusions

Screening for chitinolytic activity in 66 bacterial isolates of regions of the Pacific Ocean and several type strains from the KMM collection revealed the highly active representatives belonging to the genera *Microbulbifer* (nine), *Vibrio* (three), *Aquimarina* (two), *Pseudoalteromonas* (two), and *Zobellia* (one). The whole genome of the strains of *Microbulbifer thermotolerans* (Sgf 25, Sgm 25/1, Sh 1, Sh 2, Sh 3, Sh 4, Sh 5, 14G-22, and JCM 14709^T^), *Vibrio* spp. (Sgm 5, Sgm 22, and 14G-20), *Pseudoalteromonas* sp. B530, *Pseudoalteromonas flavipulchra* KMM 3630^T^, *Aquimarina muelleri* (V1SW 51 and V1SW 49^T^), and *Zobellia galactanivorans* Dji^T^ contain different chitin-degrading systems and enzyme numbers. The species- and strain-specific chitinase activities against colloidal chitin, which was found in the culture supernatant of *M. thermotolerans* Sgf 25 and *Z. galactanivorans* Dji^T^, were confirmed to be caused by their predicted genes annotated as the GH18 and GH23 families. The recombinant products from the GH18 of *M. thermotolerans* Sgf 25 (accession no. MCX2831407, MCX2830300, MCX2831405, and MCX2831458) and *Z. galactanivorans* Dji^T^ (accession no. CAZ95072 and CAZ97205) possessed chitin-degrading activity towards colloidal chitin. Two GH18 chitinases and one GH23 (CAZ95778) of the agarolytic *Z. galactanivorans* Dji^T^ (CAZ95072 and CAZ97205) have endo- and exochitinase activity that suggest its ability for the deep processing of red algae with a chitin-containing endoskeleton. The strains *P. flavipulchra* KMM 3630^T^, *Pseudoalteromonas* sp. B530, and *Vibrio* sp. Sgm 5 belonging to the species *V. jasicida* have one GH19 gene each in addition to GH18 (two) and LPMOs (two). This may explain the significant suppression of the *Aspergillus niger* hyphal growth at their co-cultivation; therefore, they are considered for biocontrol agents’ development in further research.

## Figures and Tables

**Figure 1 microorganisms-11-02255-f001:**
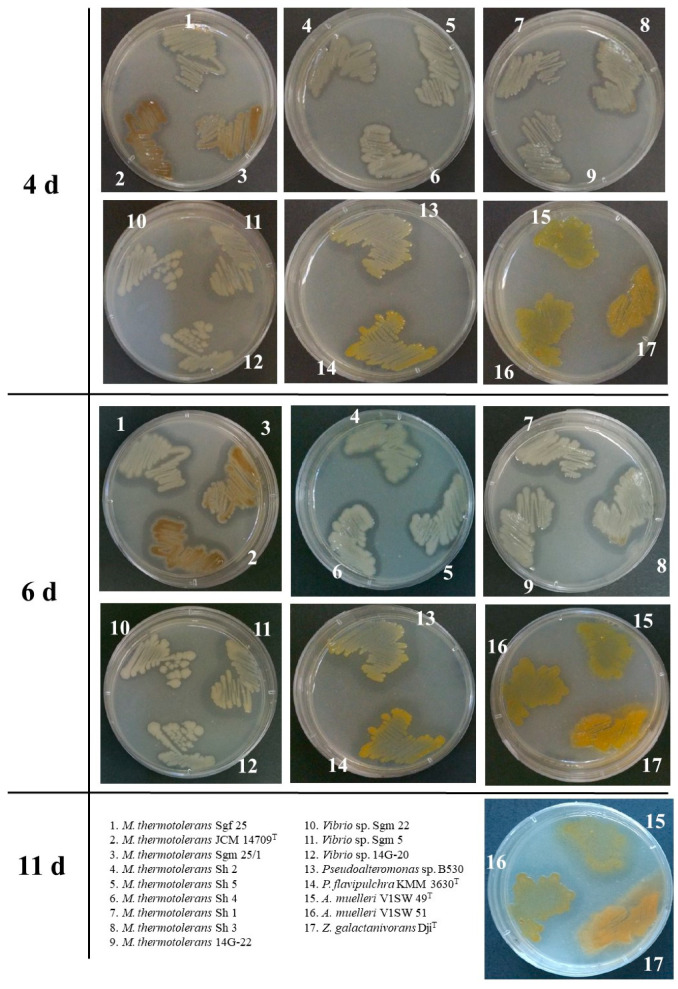
The degradation of colloidal chitin, supplemented into LB agar medium, by the marine bacterial strains: clearing zones were around colonies of *Microbulbifer thermotolerans* on the 4th–6th days; *Vibrio* sp. Sgm 5 on the 4th day; *Vibrio* sp. 14G-20 and Sgm 22 on the 6th day; both *Pseudoalteromonas* sp. B530 and *Pseudoalteromonas flavipulchra* KMM 3630^T^ on the 6th day; *Aquimarina muelleri* and *Zobellia galactanivorans* Dji^T^ on the 11th day, caused by their putative chitinase activity.

**Figure 2 microorganisms-11-02255-f002:**
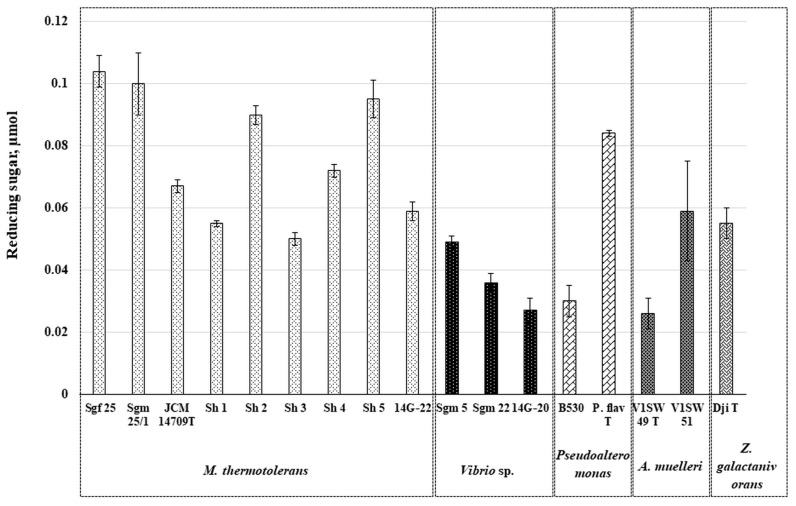
Chitinase activity in the culture supernatant of *Microbulbifer thermotolerans*, *Vibrio* spp., *Aquimarina muelleri*, and *Zobellia galactinovorans*. Reactions were carried out for 15 min, at 37 °C, using colloidal chitin (0.1%) as a substrate, and 20 mM sodium phosphate buffer, pH 6.0. The experiments were performed in triplicate and standard deviations are shown as error bars.

**Figure 3 microorganisms-11-02255-f003:**
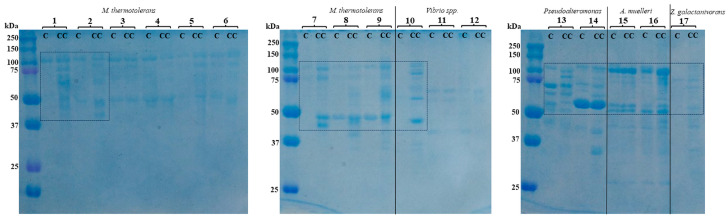
SDS–PAGE analysis of the culture supernatants: lane kDa, protein size markers (BioRad); lanes C—control culture supernatants grown without colloidal chitin, lanes CC—culture supernatants grown in the presence of colloidal chitin. The lanes under square brackets 1—*Microbulbifer thermotolerans* Sgf 25; square brackets 2–6—*M. thermotolerans* Sh 1–5; square brackets 7—*M. thermotolerans* JCM 14709^T^; square brackets 8—*M. thermotolerans* Sgm 25/1; square brackets 9—*M. thermotolerans* 14G-22; square brackets 10—*Vibrio* sp. Sgm 5; square brackets 11—*Vibrio* sp. 14G-20; square brackets 12—*Vibrio* sp. Sgm 22; square brackets 13—*Pseudoalteromonas* sp. B530; square brackets 14—*Pseudoalteromonas flavipulchra* KMM 3630^T^; square brackets 15—*Aquimarina muelleri* V1SW 51; square brackets 16—*A. muelleri* V1SW 49^T^; square brackets 17—*Zobellia galactanivorans* Dji^T^. Predicted induced chitin-degrading enzymes compared to control are underlined by squares with broken line. Protein was concentrated to 30 µg with 10% trichloroacetic acid and applied into the well.

**Figure 4 microorganisms-11-02255-f004:**
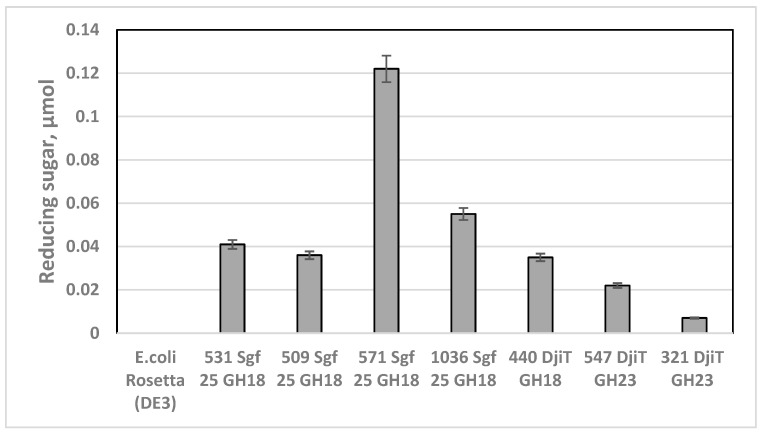
Chitinase activity of the recombinant proteins: 531 Sgf 25 GH18, 509 Sgf 25 GH18, 571 Sgf 25 GH18, and 1036 Sgf 25 GH18 from *Microbulbifer thermotolerans* Sgf 25; 440 DjiT GH18, 547 DjiT GH23, and 321 DjiT GH23 from *Zobellia galactanivorans* Dji^T^. The crude extract of *E. coli* Rosetta DE3 transformed by the plasmid pET40 b(+) without a target gene insertion was used as the negative control. The chitinolytic activity was determined by measuring the amount of reducing ends liberated from enzymatically hydrolysed colloidal chitin.

**Figure 5 microorganisms-11-02255-f005:**
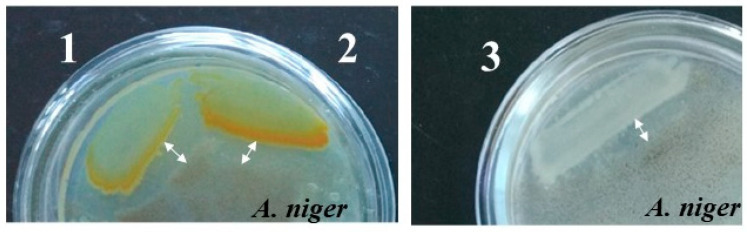
Antifungal activity of the isolates against the hyphal growth of *Aspergillus niger*. 1, *Pseudoalteromonas* sp. B530; 2, *Pseudoalteromonas flavipulchra* KMM 3630^T^; 3, *Vibrio* sp. Sgm 5. The inhibition of mycelial extension was determined by visual inspection on agar plates (Difco Marine Broth, bacteriological agar-agar, 0.2% colloidal chitin) and shown by arrows. The experiments were performed in triplicate.

**Table 2 microorganisms-11-02255-t002:** Functional confirmation of the predicted genes encoding for chitinolytic enzymes in *Microbulbifer thermotolerans* and *Zobellia galactanivorans*.

	Primers		ID GenBank			Enzyme Activity	
Gene Name *	Forward (5′-3′) (Restriction **)	Reverse (5′-3′) (Restriction, Annealing ***)		Gene Length, bp	Protein MW; Isoelectric Point	Colloidal Chitin, U/mg	4-Nitrophenyl N-Acetyl-β-D-glucosaminide, U/mg
	***Microbulbifer thermotolerans*** **Sgf 25**						
	**GH18 family**						
531 Sgf 25	tatagagctccttcgattgcagtggtttgc (Sac I)	tatactcgagttaaaggttgtcgaggaatgtcc (XhoI, 55 °C)	MCX2830300	1596	56.67 kDa 4.99 pI	0.055 ± 0.002	ND ****
509 Sgf 25	tatagagctccgtggactgcagaagtctgc (Sac I)	tatagtcgacttagggcagattgtgtacatg (Sal I, 55 °C)	MCX2831405	1530	52.78 kDa 4.37 pI	0.048 ± 0.001	0.00069 ± 0.00003
571 Sgf 25	tataccatggcgtacgattgcggaggcgtac (Nco I)	tatagagctcctactgattactgtgcatggcttc (Sac I, 55 °C)	MCX2831407	1716	59.91 kDa 4.38 pI	0.163 ± 0.004	0.00023 ± 0.00001
1036 Sgf 25	tatagagctcctacgattgcagcgggctg (Sac I)	tatactcgagttaattcgctgagcaaatgagtgtc (XhoI, 62 °C)	MCX2831458	3111	111.3 kDa 4.28 pI	0.073 ± 0.002	ND
	***Zobellia galactanivorans*** **Dji^T^**						
	**GH18 family**						
527 Dji^T^	tataccatggaggaacaaagtttggatcagg (NcoI)	tatagagctcttaattacaagaccgtaccacc (SacI, 72 °C)	CAZ95072.1	1584	56.63 kDa 4.35 pI	ND	0.00140 ± 0.00006
440 Dji^T^	tataccatggcgagtgatgatactaattattcttcacc (NcoI)	tatagagctcctaattaaactgattaagtatatc (SacI, 45 °C)	CAZ97205.1	1323	49.41 kDa 4.46 pI	0.047 ± 0.002	0.00113 ± 0.00004
	**GH23 family**						
547 Dji^T^	tataccatggcgcaagaacaggaaagagattctg (NcoI)	tatactcgagttacgatgaacattcgcac (XhoI, 50 °C)	CAZ95778.1	1644	62.25 kDa 8.23 pI	0.030 ± 0.001	0.00078 ± 0.00002
321 Dji^T^	tataccatggtacaaaataatgtagaaccgag (NcoI)	tatagagctcctacttacctaccctataatac (SacI, 50 °C)	CAZ94149.1	966	36.73 kDa 5.15 pI	0.009 ± 0.001	0.00080 ± 0.00003

* Sequence number of the orthologous gene in the chromosome of *M. thermotolerans* DAU221 (CP014864) and *Z. galactanivorans* Dji^T^ (FP476056) according to the CAZy database; ** the endonuclease restriction sites are in underlined; *** annealing temperature of the PCR performed at the conditions recommended by the Q5^®^ High-Fidelity DNA Polymerase’s instruction (NEB, USA); ND ****—not determined.

## Data Availability

The datasets presented in this study can be found in online repositories. The names of the repository/repositories and accession number(s) can be found in the article/Appendix A.

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
