# Peer review of "Chitinolytic and Fungicidal Potential of the Marine Bacterial Strains Habituating Pacific Ocean Regions"

_microorganisms, 2023, doi:10.3390/microorganisms11092255_

Round 1

Reviewer 1 Report

The main issues in this article include:

1:FIgure 5 provided bacterial inhibitory activity against Aspergillus niger, the main problem with this result is that bacteria and fungi have spatial competition, and in many cases bacteria with no antifungal activity can exhibit antimicrobial circles. Therefore in most cases it is also necessary to use supernatants to corroborate the bacterial inhibition experiments.

2:Figure 4 This result lacks a negative control for the crude extract of DE3 supernatant, and there is no need to use a stereoscopic presentation of the morphology in the picture, which could easily lead the reader to misunderstand that there is another parameter

In addition there are numerous writing errors in the article, such as Latin names not being italicised, the first occurrence of a Latin name in a figure legend should give the full name,figure 2 repeats , etc. 

Reviewer 2 Report

The manuscript by Pentekhina et al. investigated chitinolytic and antifungal activity of diverse bacterial strains isolated from marine environment. Chitinase encoding genes were identified by genome sequencing of these bacteria. The function/activity of some of the identified chitinase genes were confirmed by heterologous expression of the chitinase gene in E. coli and followed by chitinase activity assay using the cell extract. Overall the study is well designed and well done. However, the text and data presentation and interpretation should be more clear and focused. Please consider the followings for further improvement.

1.   A table with the 17 characterized bacterial strains need to be provided as a main table in the manuscript, and information such as their chitinolytic and antifungal activity, presence/numbers and type of chitinase genes in their genome should also be added in the table. Such table can provide a clear guidance for the readers of this paper.

2.   Line16-21 in the Abstract is useless.

3.   Line22. What are the 17 isolates? Where are they from? Why they are selected in this study? What did you do with these isolates? And what did you find? Information regarding these are needed.

4.   Line 40,41. Do “nature” and “environment” here mean marine environment? If yes, better replace them with marine environment. As “marine bacteria” is specified in line41.

5.   Line70. Remove the comma between “enzymes” and “have”.

6.   Line111, three GH18 chitinases?

7.   Line188. Remove “A total”

8.   Line194. “Crude enzyme” is the protein extracted from the supernatant, right? Better explain a little more here for clarification.

9.   Line228-230. So the chitinase protein expressed in E. coli was intracellular? Is this because E. coli doesn’t have the transport system to export the chitinase from the cell?

10.  Line236. What is the protein concentration in the 50 ul E. coli cell extract?

11.  Line346, 353. The unit of genome size is probably wrong. kbp, rather than bp?

12.  Figure 3. What are these bands/proteins stand for? Which ones are the chitinase? This should be indicated in the SDS-PAGE gel picture. Otherwise people cannot know the expression level (Line458).

13.  Figure 2 and 4. Were the same substrate used for measuring chitinase activity? Why not using the same unit (U) for the two graphs? And it’s not necessary to repeat the enzyme assay method in the figure legend.

14.  Error bar is missing in figure 4.

15.  Line434. How many chitin-degrading enzymes are encoded in the strains of M. thermotolerans?

16.  Line612. Other Vibrio strains (14G-20, Sgm22) also have GH19 enzyme encoding genes. Do they have such antifungal activity? If not, why?

17.  The Result and Discussion section contains huge amount information about taxonomic classification of the bacterial strains used in the study. Why this is so important? Didn’t these strains already well identified/classified before they were deposited to the culture collection center?

18.  A Conclusion section is needed, especially given that the manuscript contains dense information about the physiology, genetics, and biochemistry of diverse bacterial strains.

Round 2

Reviewer 1 Report

This article can be published in its current form.

Reviewer 2 Report

The manuscript by Pentekhina et al. has been nicely revised and the previous issues and reviews' comments are also well addressed. I don't have more questions/comments.